:ΦPLOS | ONE

# The impact of language on the interpretation of resuscitation clinical care plans by doctors. A mixed methods study

Colette Dignam[1,2]*, Josephine Thomas[3], Margaret Brown[4], Campbell H. Thompson[1,2]

1 Division of Medicine, Faculty of Health and Medical Sciences, University of Adelaide, Adelaide, South Australia, Australia, 2 Division of Medicine, Royal Adelaide Hospital, Adelaide, South Australia, Australia, 3 Adelaide Medical School, Faculty of Health and Medical Sciences, University of Adelaide, Adelaide, South Australia, Australia, 4 School of Psychology, Social Work, and Social Policy, Division of Education, Arts and Social Sciences, University of South Australia, Adelaide, South Australia, Australia

☯ These authors contributed equally to this work.
* colette.dignam@sa.gov.au

**Data Availability Statement:** Data cannot be shared publicly because permission was not obtained from survey participants to share their answers in a public database and due to the single

## Abstract

### Introduction

Resuscitation clinical care plans (resuscitation plans) are gradually replacing 'Not for Cardiopulmonary Resuscitation' orders in the hospital setting. The 7-Step Pathway Resuscitation Plan and Alert form (7-Step form) is one example of a resuscitation plan. Treatment recommendations in resuscitation plans currently lack standardised language, creating potential for misinterpretation and patient harm.

### Aims

To explore how terminology used in resuscitation plans is interpreted and applied by clinicians.

### Method

A mixed methods study surveyed 50 general medical doctors, who were required to interpret and apply a 7-Step form in three case vignettes and define seven key terms. Statistical analysis on multiple choice and thematic analysis on free-text responses was performed.

### Results

Terminology was inconsistently interpreted and inconsistently applied, resulting in clinically significant differences in treatment choices. Three key themes influenced the application of a resuscitation plan: in-depth discussion, precise documentation and personal experience of the bedside deciding doctor.

centre nature of the study, the ethics committee were concerned about inadvertent identification of participants. This is in accordance with the current ethics approval. Permission to review the data can be requested from the Royal Adelaide Hospital ethics committee (contact via health. CALHNresearchLNR@sa.gov.au) for researchers who meet the criteria to review the data.

**Funding:** The authors received no specific funding for this work.

**Competing interests:** The authors have declared that no competing interests exist.

## Discussion

This study highlights persistent communication deficiencies in resuscitation plan documentation and how this may adversely affect patient care; findings unlikely to be unique to Australia or South Australia.

## Conclusion

Removing ambiguity by standardising and defining the terminology in resuscitation plans will improve bedside decision-making, while also supporting the rights of the patient to receive appropriate and desired care.

## Introduction

Clear documentation of medical treatment goals on a dedicated form helps patients receive appropriate and desired care, particularly in the setting of unexpected deterioration. Despite its recognised importance, there is no internationally accepted title for this type of documentation; terms in use include goals of care plan, resuscitation clinical care plan (resuscitation plan), and emergency treatment plan [1–3]. For the purposes of this study, we define a resuscitation plan as a document written by a doctor responsible for a patient's care which includes a directive regarding cardiopulmonary resuscitation (CPR), but also contains recommendations and/or limitations about other interventions, such as intubation or use of life-prolonging antibiotics. Documentation should record the goals of care following a discussion with the patient or substitute decision-maker (SDM) about the treatment preferences, informed by the doctor's medical assessment of likely treatment outcomes [1,2].

In 2014, the 7-Step Pathway Resuscitation Plan and Alert form (7-Step form) (S1 Fig) was introduced as an integral part of implementing the *Advance Care Directives Act 2013* (SA) to provide a clinical tool to reinforce the principle of self-determination; the individual's right to refuse treatment including CPR. This resuscitation plan is now used in all six South Australian urban public referral hospitals and is available in smaller rural hospitals and clinics staffed by primary care providers, it is also recognised as a treatment guide by ambulance officers [3]. The form and underlying Resuscitation Planning Policy Framework were developed to address previously identified inadequacies in the process of resuscitation decisions and documentation, with a renewed emphasis on transparent patient-doctor discussion that would result in desired and appropriate medical care, while protecting the right to withhold futile interventions.

The completed form represents the final step in a process of creating a care plan in conjunction with the patient; a stepwise approach that is locally known as the 7-Step Pathway. In accordance with current evidence base, the 7-Step Pathway includes medical assessment, open consultation with the patient about their treatment goals, and a transparent discussion of appropriate treatment within the framework of known co-morbidity and quality of life [3,4]. The plan should be communicated to the primary care provider on hospital discharge, and a carbon-copy provided to the patient. While the 7-Step form is not legally binding in isolation, it may be used to reflect previously documented binding treatment refusals that a patient has recorded on an Advance Care Directive.

In line with this study's definition, international examples of resuscitation plans also contain a CPR directive, a description of treatment limitations or recommendations, and a record

of patient/SDM involvement. Key differences between examples include the relative focus on free text documentation vs standardised tick box care options, whether the document is used for a specified length of time or indefinitely (as is the case with the 7-Step form), and the involvement of senior staff in creating a plan [1,5,6].

Previous analysis has shown that when compared to ad hoc case-note documentation, the 7-Step form improved frequency of resuscitation documentation, but did not reduce the use of undefined and ambiguous terminology [4]. In addition to the lack of a common title, international resuscitation plans include ambiguous or undefined terms including 'ward care', 'supportive care' and 'limited intervention' [7–9]. The lack of standardised language may create a risk of inappropriate patient care through misinterpretation.

This study investigates whether terminology used in a resuscitation plan is both consistently interpreted and consistently applied by a clinician at the bedside. Secondary aims were to determine whether clinicians found a standardised resuscitation plan useful, and to explore the other factors which influence how they applied a resuscitation plan.

## Method

A mixed methods approach was designed to maximise insight into resuscitation plan interpretation. A closed internet survey, accessed via emailed link, was sent to all one hundred doctors who had worked in general medicine at a single centre between February-August 2017. The survey was conducted between August and October 2017. The email and link stated the purpose, duration and nature of the survey and that completion of the survey was taken for consent, in accordance with ethics approval (Royal Adelaide Hospital: HREC/17/RAH/375). Participation was anonymous, voluntary and no incentive was provided. Survey questions were pilot tested with the co-authors (MB, CHT) and physician colleagues, and refined iteratively. Questions were not randomised. The Checklist for Reporting Results of Internet E-Surveys was used as a reporting guide for this study [10].

In the first section of the survey, three case vignettes drawn from real life examples were described (S1 Text). Having read the resuscitation plan, participants selected all interventions they deemed appropriate from a list of multiple-choice options. In the second section, participants defined seven terms used in the free-text section of the 7-Step form; identified from previous research [4]. The final section was a Likert Scale response to the statement 'the 7-Step Pathway aids decision-making about appropriate goals of care, not just CPR', with options ranging from strongly agree to strongly disagree. Participants could expand on all responses with free-text.

The Fisher-exact test was used to compare the treatment choices of senior and junior doctors and differences were highlighted where statistical significance was reached (p = 0.05). In each case vignette, responses were also ranked from most consistently chosen to least consistently chosen with 100% or 0% representing maximum agreement between respondents, and 50% representing maximum disagreement. The ranked responses were used to calculate a standard deviation that quantified the relative level of agreement between respondents' answers in each case. Standard deviations of the three case vignettes were compared using Lehmans' variance ratio to determine statistical significance (p = 0.05).

Thematic analysis was performed on the free-text and semi-structured responses using the established methodology of Braun and Clark [11]. Each respondent was assigned a unique identifier including a number and a letter (C = Consultant, R = Registrar, M = Resident Medical Officer, I = Intern). Following data familiarisation, preliminary concepts from each participant's responses were identified and coded inductively. Themes and subthemes were sought until all data was reviewed or saturation reached. The second author who is a local expert with

experience in both the subject matter and qualitative research independently reviewed the data to ensure all themes were adequately captured and no themes or divergent views were missed. Links between free-text responses and multiple-choice treatment selections were sought and new themes developed or incorporated into existing themes where appropriate.

## Results

### Characteristics of participants

Fifty responses from the one hundred invited participants were obtained (50%), predominantly from interns and residents (Table 1). This is consistent with the General Medicine ward structure where the junior staff rotate every three months and therefore total numbers are larger. 31/50 (62%) participants had completed education about the 7-Step form, chiefly via attendance at in-hospital presentations. There was no statistical difference in the prevalence of those who received education across the levels of seniority.

Overall, 39/50 (78%) of doctors agreed or strongly agreed that the 7-Step form aids decision-making about appropriate goals of care, not just CPR. There was no association between education received and the clinicians' opinion of its usefulness (p = 0.9). 87% of interns agreed the 7-Step form is a useful decision aid, compared to 57% of consultants. The difference did not reach statistical significance (p = 0.08).

### Case vignette interpretation

The abridged case vignettes are outlined in Table 2, and the treatment choices selected in response to each case vignette shown in Fig 1A, 1B and 1C.

The treatment choices selected in response to each case vignette are shown in Fig 1A, 1B and 1C. In Case #1-Comfort Care, 96% of respondents selected subcutaneous morphine and did not select intravenous fluid bolus, chest X-ray or arterial blood gas. Consultants were more likely than junior doctors to select intravenous antibiotics, chest x-ray and arterial blood gas (p = 0.04 in all three examples). Supplemental oxygen using nasal specs and non-rebreather were selected by 72% and 18% respectively. It was unclear from free-text responses if oxygen was given for symptomatic relief or to extend life.

In Case#2-No Life Prolonging Treatment, morphine and midazolam were selected by 82% and 85% of responders respectively. Potentially life-prolonging options were selected by 26% of respondents, including intravenous fluid (26%), intravenous antibiotics (24%) and oral antibiotics (14%). 42% of consultants prescribed intravenous antibiotics versus 23% of junior staff, though the difference was not statistically significance (p = 0.3).

In Case#3-Ward Measures, more than 90% of respondents selected the options of intravenous fluid bolus, electrolyte replacement and cardiology consult. Intravenous atropine, coronary care unit admission and external cardiac pacing were selected by 78%, 66% and 50% of respondents respectively. Consultants were more likely to give isoprenaline and external pacing than junior doctors (p = 0.03).

**Table 1. Characteristics of participants.**

|  | Intern | Resident | Registrar | Consultant | TOTAL |
|---|---|---|---|---|---|
| Participants (% of invitees) | 23 (48) | 17 (52) | 3 (38) | 7 (37) | 50 (50) |
| Had received education (%) | 13/23 (57) | 11/17 (65) | 2/3 (67) | 5/7 (71) | 31/50 (62) |
| Agreed[a] (%) | 20/23 (87) | 13/17(76) | 2/3 (67) | 4/7 (57) | 39/50 (78) |

[a] These doctors agreed or strongly agreed with the statement 'The 7-Step Pathway aids decision-making about appropriate goals of care, not just CPR'

**Table 2. Abridged case vignettes (See S1 Text for unabridged case vignettes).**

| |
|---|
| Case #1-Comfort Care: A dyspnoeic patient with End-Stage Chronic Obstructive Pulmonary Disease has deteriorated despite maximum therapy. The 7-Step form tick-box section states 'Not for any Treatment Aimed at Prolonging Life'. The free-text states 'Comfort care'. |
| Case #2 -No Life Prolonging Treatment: A patient with end-stage dementia presents with likely urosepsis. The patient's Advance Care Directive states 'If my dementia gets worse and I can no longer make my own decisions, I do not want any treatment to sustain life.' The 7-Step form tick-box section states 'Not for any Treatment Aimed at Prolonging Life/Not for MER (Medical Emergency Response) calls'. The free-text is blank. |
| Case #3- Ward Measures: A patient is reviewed at a MER call for profound drowsiness, hypotension and heart rate 30bpm. She has a history of atrial fibrillation on metoprolol 50mg twice daily, and quiescent metastatic breast cancer. A 7-Step form was previously completed with her husband. The tick-box section states 'Not For CPR/ invasive ventilation/intensive care treatment or admission'. The free-text states 'Active ward measures'. |

The quantified level of agreement between clinicians about appropriate treatment was greatest in Case#1, then Case #2, then Case #3 with standard deviations of 10.8, 11.7 and 19.3 respectively. The difference in level of agreement between Case #1 and Case #3 was statistically significant (p<0.001), suggesting that 'comfort care' results in a more consistent approach to management than 'active ward measures'.

## Definitions of key terms in resuscitation plans

Forty-eight doctors completed this section. 29 (60%) participants used one repeated definition for all five 'ward-' terms or wrote 'as above', indicating they deemed these terms synonymous. For respondents who gave multiple definitions, the intent of treatment and the perceived appropriate limit of care varied significantly, as seen in Table 3. 'Active measures' and 'full ward measures' were associated with the provision of more aggressive treatments, including CPR for 20% of respondents.

For 'comfort care' and 'palliative approach', goals of care were reflected in language like 'symptomatic relief', 'dying' and 'alleviate distress', with 31/47 (65%) of respondents using a single definition for both terms. In responses where the terms were separately defined, the chronology between 'palliative approach', 'comfort care' and imminent death varied. Opinions varied about whether artificial hydration, antibiotics and chemotherapy were part of a 'palliative approach', especially in patients presenting with an illness other than their underlying life-limiting condition. Of the seven terms, 'comfort care' was the term where intent of care was the most concordant.

## Thematic analysis

Analysis of free-text responses revealed three key themes that influenced how a resuscitation plan is interpreted and applied. These were 'the discussion', 'the documentation' and 'the deciding doctor'.

The theme 'the discussion' pertains to the communication between the documenting clinician and patient/SDM prior to completion of a resuscitation plan. A robust and clearly documented discussion was valued by respondents, and considered pivotal to implementing the clinical care plan. When the 7-Step form was incomplete or there was inadequate supporting documentation describing the discussion, respondents acknowledged misinterpretation was possible.

In Case#3-Ward Measures, participants perceived that inadequate discussion had occurred and expressed reduced confidence in the documented resuscitation plan. They indicated a reduced willingness to make clinical decisions without additional information. Participants suggested that when the initial discussion was not sufficiently detailed, workload and patient

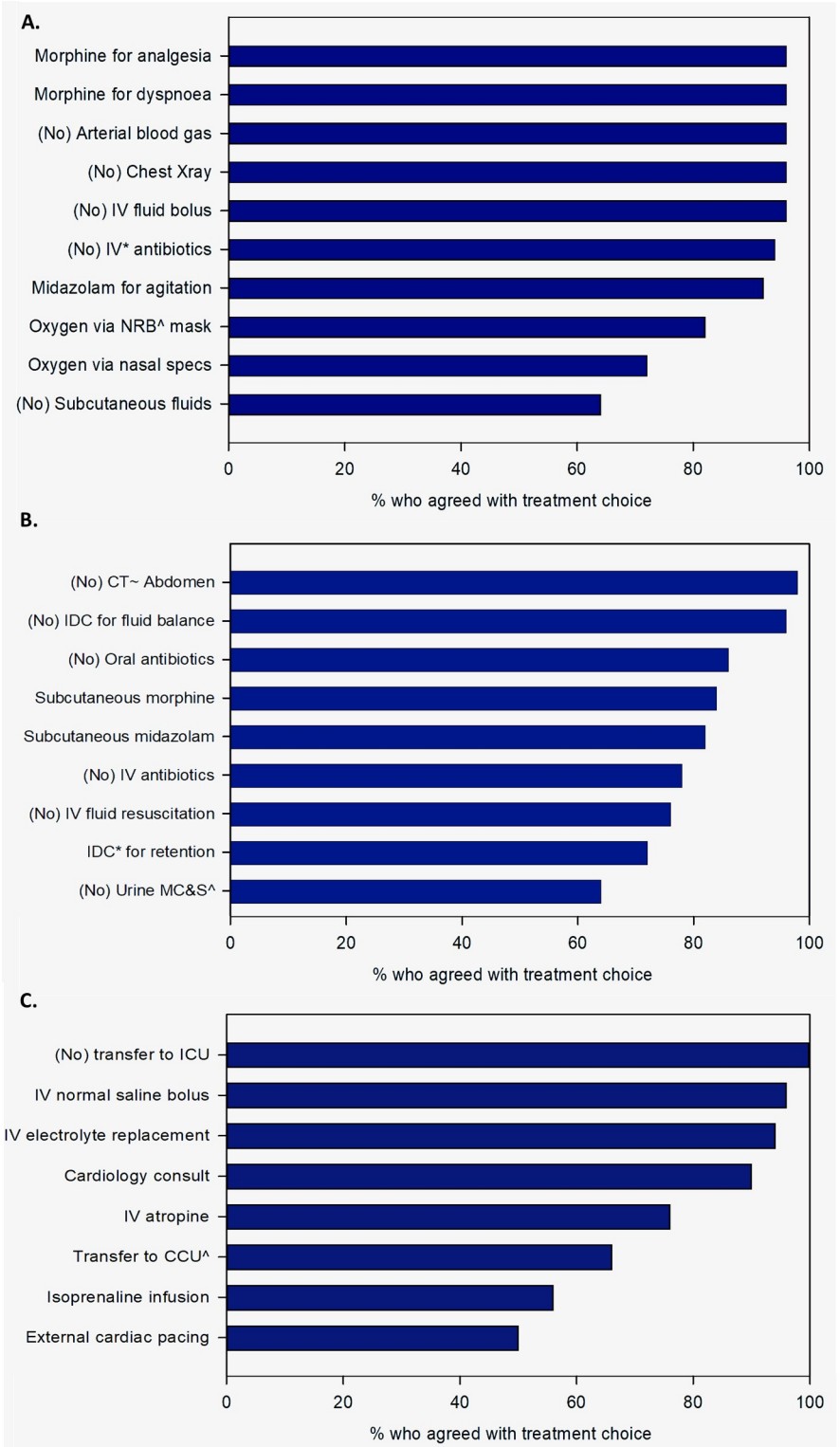

**Fig 1. Percentage of respondents who agreed with each treatment option in each case vignette.** (A) Case #1-
Comfort care: [+]Where a treatment was selected by <50%, the inverse proportion has been displayed in order to
demonstrate degree of concordance between doctors i.e. 6% selected antibiotics so 94% selected (No) antibiotics.
^Non-rebreather. *Intravenous. (B) Case #2- No life prolonging treatment: [+]As in (A), the inverse proportion has been
graphically displayed. ^Microscopy, sensitivities and culture. *In-dwelling urinary catheter. ~Computerised

Tomography scan (C) Case #3- Ward measures: [+]As in (A), the inverse proportion has been graphically displayed. [^]Coronary Care Unit.

care were both adversely affected. A pre-emptive or early discussion was preferred to impromptu discussion prompted by patient deterioration.

> "I have been [in] situations before where 7-Steps have been misinterpreted as that relevant box has not been discussed with the patient." (I6)

> "Every day multiple METs occur where someone in T2RF [Type 2 Respiratory Failure] is "Not for CPR/ICU/[intubation]" but there is no mention of NIV [Non-invasive ventilation] . . . resulting in multiple phone calls to determine the true ceiling of care" (M15)

The theme, 'the documentation', encompasses the available written information at the bedside in both the 7-Step form and/or case notes. The change to a standardised resuscitation plan was viewed positively because of ease of locating and increased clarity, though not all respondents felt it had changed their practice.

The form was reported to clearly indicate if a patient should not receive CPR, and when the focus was strictly palliative. However, respondents were critical of how the free-text section was used; ill-defined or general statements were reportedly common and rarely added clarification to treatment goals. In particular, the phrase 'ward measures' was perceived as a source of frustration and ambiguity. Since treatments available in each ward vary, it was unclear to the respondents what treatments were actually appropriate. The addition of other specific treatments (e.g. non-invasive ventilation) to the tick-box section was suggested to clarify this uncertainty.

> "The 7-step pathway has not changed how I document my resuscitation plans or how I have the discussion with patients and families. It has provided a common place to write the resuscitation orders in the case notes." (R2)

> "There should be also a move to further specify treatments that are appropriate/inappropriate as many 7-steps end up with very vague and generalised statements." (M14)

**Table 3. Selected definitions of key terms.**

| Key term to define | Selected definitions |
|---|---|
| Full ward measures | "Everything possible, including resuscitation" (M4)<br>"Everything except ICU/CPR/intubation" (M6) |
| Active ward measures | "I find this unclear-unsure if it would include escalation to ICU" (M2) |
| Ward based measures | "An intervention which can be safely performed on a ward; not necessarily to prolong life" (I21) |
| Ward based management | "Treatment outside of ICU, aim to cure/prolong life. May include a procedure depending on the situation." (C2) |
| Ward based care | "Measures which can be given on a ward but which do not actively pursue a new diagnosis" (I10) |
| Comfort Care | "Aimed at allowing the patient to die with dignity with relief from symptoms such as pain, dyspnoea, agitation." (I9) |
| Palliative Approach | "This is a very ambiguous term, palliative approach can mean ensuring best quality of life. . .thus may also involve some active management" (C4)<br>"Comfort care for patient certain to die" (C5) |

"Ward based measures" is okay, but fails to adequately identify the ceiling limit of care. Lots of therapies can take place on the wards, especially on "specialty wards." (M15)

The theme, 'the deciding doctor', acknowledges the bedside clinician as an individual, whose treatment decisions are affected by personal values and experience. Interns were unlikely to question the resuscitation plan's instructions and valued specific limitations that could be readily followed. For more senior doctors, there were conflicting views of the resuscitation plan's role in determining care, ranging from a useful tool to a guideline that stood in the way of individualised patient assessment.

The deciding doctors' treatment choices were therefore impacted by the value they placed on the resuscitation plan and its specified treatment limitations, with some doctors willing to overrule the plan (or at least re-open the discussion). This concept was evident in the free text responses and the treatment selections in case vignettes.

"[The] decision has to be made on individual circumstances on the day, protocols can be a help or a distraction." (C5)

"As with anything, the more information provided by the person who is writing the 7-step, the more useful it is to determine the appropriate goals of care." (I21)

Re Case #3: "In this case I'd probably ignore the 7-steps if what was discussed wasn't documented, and I'd call the patients (sic) consultant too." (M8)

## Discussion

Despite resuscitation plans being regarded as a useful tool by 78% of respondents, this study shows that much of the terminology used in the 7-Step form is inconsistently interpreted and inconsistently applied by doctors, resulting in clinically significant differences in treatment choices.

Responses to all of the case vignettes showed treatment inconsistencies, with Case#1-Comfort care showing greatest agreement about appropriate management. Despite the highest level of agreement and a narrow range of definitions for 'comfort care', there was still inconsistent provision of oxygen and fluids by respondents in the case vignette. This finding is similar to a USA study where 176 surveyed clinicians had disparate opinions about whether 'comfort measures only' included intravenous antibiotics, supplemental oxygen or emergency response calls [12].

In Case #2-No Life Prolonging Treatment, 24% of respondents prescribed potentially life-saving intravenous antibiotics for a septic patient whose binding Advance Care Directive stated 'No treatment to sustain life'. Prescribing antibiotics is at best a problematic misinterpretation, making clear the need for standardised definitions. It could also be a paternalistic disregard for a patient's right to refuse medical treatment. This is not an isolated occurrence; one USA study found 2–13% of surveyed doctors would provide CPR for patients with a 'Do Not Resuscitate' order [13].

In Case #3-Ward measures, uncertainty was greatest about whether sub-specialty emergency interventions like isoprenaline infusion and external pacing were appropriate. Uncertainty was also evident in the range of definitions given for each of the five 'ward-' terms: while 60% of respondents would assume the same goals of care regardless of whether 'full ward measures' or 'ward based care' was documented, the remaining 40% of doctors thought each term reflected a different treatment approach, including whether they should aim to prolong life.

Thematic analysis revealed that the implementation of a resuscitation plan is confounded by factors other than undefined language. Confidence in a resuscitation plan was greatest when there was a robust and clearly documented discussion with adequately detailed treatment limitations. Even when adequately detailed, the bedside deciding doctor may re-discuss, ignore, or flexibly interpret a resuscitation plan they disagreed with. Interns, who are inexperienced in clinical decision-making and are used to following senior orders, were also more likely to both closely follow a resuscitation plan and consider it a useful decision-making tool. This may in part explain the statistical differences in treatment approaches seen in the case vignettes. International studies have similarly shown that resuscitation plans are relied upon by after-hours junior doctors to accurately convey the home team's treatment recommendations, and improve speed of decision-making [14,15]. Many international resuscitation plans require a secondary signature from a senior officer, which may be one practical solution to improve cohesion in the approaches of junior and senior staff.

While not the focus of this study, the results suggest that the perceived purpose of a resuscitation plan varies between doctors: is it predominantly a handover tool, or a document to protect patients from intervention they do not want? These objectives should ideally align, but translation of a patient's wishes into documented treatment limitations is an imprecise process. Misinterpreting a resuscitation plan based on an Advance Care Directive, or relying on poor quality documentation, does not simply represent a failed medical handover however. It represents a loss of the patient's voice in the decision-making process. The underlying aim of resuscitation planning policy, to champion the patient's voice, has arguably not been achieved [16].

An ideal resuscitation plan therefore needs to strike a balance between patient advocacy, a clear medical handover, and the unpredictable nature of future hospital admissions. While junior doctors in this study were in favour of increased standardised choices such as 'Not for non-invasive ventilation', a list of this nature will inevitably be incomplete and maintains a focus on the treatment that will be withheld. Contemporary alternatives include the model exemplified in the USA-based POLST (physician's orders for life sustaining treatment) form, where patients are divided into three broad treatment approaches; full, selective and comfort-focussed [17]. The increased clarity and ease of use in this three-tiered care model potentially comes at the expense of a nuanced and individualised patient-centred approach. The ReSPECT (Recommended Summary Plan for Emergency Care and Treatment) model in United Kingdom provides ample scope for individualisation and patient involvement in the form of visual scales balancing curative and symptom-based care [5]. This form requires the doctor to accurately document treatment limitations in free-text format however, thereby risking the use of undefined or ambiguous terminology that proved so problematic in this study.

This study is limited to a single-centre general medical unit with 50% response rate which may limit generalisability of findings. A strength of the selected cohort is that this general medical unit completes resuscitation plan forms on 63% of patients over 70 years, and has been using the 7-Step form in its current format since 2014. Sampling from this cohort therefore, removes confounding factors such as form variability or human error due to unfamiliarity with the form. The online format of the survey may have resulted in less comprehensive responses than verbal interviews.

In addition to on-site education at the time of its launch at each new site, training about the 7-Step Pathway is available through online educational modules. The adequacy of these interventions to support the completion and interpretation of the form is a confounding factor that has not been addressed in this study. The capacity for a renewed education strategy and dedicated resources to improve the process, as opposed to changing the form, remains an area for future research.

## Conclusion

Terminology in resuscitation plans is neither consistently interpreted nor consistently applied, which affects bedside treatment decisions. The deciding doctor is also influenced by personal experience, seniority, and the perceived quality of discussions with the patient/SDM. Ambiguous language in resuscitation plans undermines a key tenet of resuscitation documentation, to record a patient's preferences in a manner that will be accurately interpreted and faithfully upheld. Moving towards universally shared and defined terminology in resuscitation plans will improve medical handover, increase physician confidence in the value of the documentation and ultimately, support delivery of appropriate and desired care for the patient.

## Supporting information

**S1 Fig. Copy of 7 step pathway resuscitation plan and alert form.**
(DOCX)

**S1 Text. Unabridged case vignettes.**
(DOCX)

## Acknowledgments

The authors thank Paul Hakendorf for his statistical expertise.

## Author Contributions

**Conceptualization:** Colette Dignam, Margaret Brown, Campbell H. Thompson.

**Formal analysis:** Colette Dignam, Josephine Thomas, Margaret Brown.

**Investigation:** Colette Dignam.

**Methodology:** Colette Dignam, Josephine Thomas, Margaret Brown, Campbell H. Thompson.

**Project administration:** Colette Dignam, Campbell H. Thompson.

**Supervision:** Josephine Thomas, Margaret Brown, Campbell H. Thompson.

**Writing – original draft:** Colette Dignam.

**Writing – review & editing:** Colette Dignam, Josephine Thomas, Margaret Brown, Campbell H. Thompson.

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
