## [Decision Letter · Decision Letter 0]

5 Sep 2019

PONE-D-19-20806

De-coding the language; how clinicians interpret resuscitation clinical care plans. A mixed methods study

PLOS ONE

Thank you for submitting your manuscript to PLOS ONE. After careful consideration, we feel that it has merit but does not fully meet PLOS ONE’s publication criteria as it currently stands. Therefore, we invite you to submit a revised version of the manuscript that addresses the points raised during the review process.

We would appreciate receiving your revised manuscript by Oct 20 2019 11:59PM. To enhance the reproducibility of your results, we recommend that if applicable you deposit your laboratory protocols in protocols.io, where a protocol can be assigned its own identifier (DOI) such that it can be cited independently in the future. For instructions see: http://journals.plos.org/plosone/s/submission-guidelines#loc-laboratory-protocols

We look forward to receiving your revised manuscript.

Kind regards,

Andrew Carl Miller

Academic Editor

PLOS ONE

Journal Requirements:

2. Please provide additional details regarding participant consent.

In the Methods section, please ensure that you have specified (i) whether consent was informed and (ii) what type you obtained (for instance, written or verbal).

If your study included minors, state whether you obtained consent from parents or guardians.

If the need for consent was waived by the ethics committee, please include this information

Additional Editor Comments:

Please ensure your work adheres to the following guideline and provide citation: Eysenbach, G. (2004). Improving the quality of web surveys: the checklist for reporting results of internet e‐surveys (cherries). Journal of medical Internet research, 6(3)e34 doi:10.2196/jmir.6.3.e34 http://www.jmir.org/2004/3/e34/

Describe the informed consent process. Where were the participants told the length of time of the survey, which data were stored and where and for how long, who the investigator was, and the purpose of the study?

If any personal information was collected or stored, describe what mechanisms were used to protect unauthorized access.

State how the survey was developed, including whether the usability and technical functionality of the electronic questionnaire had been tested before fielding the questionnaire.

Please verify that this was a “closed survey”: An “open survey” is a survey open for each visitor of a site, while a closed survey is only open to a sample which the investigator knows (password-protected survey).

In “closed” (non-open) surveys, users need to login first and it is easier to prevent duplicate entries from the same user. Describe how this was done. For example, was the survey never displayed a second time once the user had filled it in, or was the username stored together with the survey results and later eliminated? If the latter, which entries were kept for analysis (eg, the first entry or the most recent)?

How/where was the survey announced or advertised? Some examples are offline media (newspapers), or online (mailing lists – If yes, which ones?) or banner ads (Where were these banner ads posted and what did they look like?). It is important to know the wording of the announcement as it will heavily influence who chooses to participate. Ideally the survey announcement should be published as an appendix.

For the e-mail survey, were the responses entered manually into a database, or was there an automatic method for capturing responses?

Were any incentives offered (eg, monetary, prizes, or non-monetary incentives such as an offer to provide the survey results)?

Randomization of items: To prevent biases items can be randomized or alternated.

Was adaptive questioning used? With adaptive questioning (certain items, or only conditionally displayed based on responses to other items) to reduce the number and complexity of the questions.

What was the number of questionnaire items per page? The number of items is an important factor for the completion rate.

Over how many pages was the questionnaire distributed? The number of items is an important factor for the completion rate.

It is technically possible to do consistency or completeness checks before the questionnaire is submitted. Was this done, and if “yes”, how (usually JAVAScript)? An alternative is to check for completeness after the questionnaire has been submitted (and highlight mandatory items). If this has been done, it should be reported. All items should provide a non-response option such as “not applicable” or “rather not say”, and selection of one response option should be enforced.

State whether respondents were able to review and change their answers (eg, through a Back button or a Review step which displays a summary of the responses and asks respondents if they are correct).

How did you determined a unique visitor. There are different techniques available, based on IP addresses or cookies or both.

-- Cookies: If so, mention the page on which the cookie was set and read, and how long the cookie was valid. Were duplicate entries avoided by preventing users access to the survey twice; or were duplicate database entries having the same user ID eliminated before analysis? In the latter case, which entries were kept for analysis (eg, the first entry or the most recent)?

-- IP Address: Indicate whether the IP address of the client computer was used to identify potential duplicate entries from the same user. If so, mention the period of time for which no two entries from the same IP address were allowed (eg, 24 hours). Were duplicate entries avoided by preventing users with the same IP address access to the survey twice; or were duplicate database entries having the same IP address within a given period of time eliminated before analysis? If the latter, which entries were kept for analysis (eg, the first entry or the most recent)?

Was log file analysis used?: Indicate whether other techniques to analyze the log file for identification of multiple entries were used. If so, please describe.

Provide view rate (Ratio unique site visitors/unique survey visitors) if possible: Requires counting unique site visitors (not page views!) divided by the number of unique visitors of the first page of the survey. It is not unusual to have view rates of less than 0.1 % if the survey is voluntary.

Provide participation rate (Ratio unique survey page visitors/agreed to participate) if possible: Count the unique number of visitors who visit the first page of the survey (or the informed consent page, if present) divided by the number of people who filled in the first survey page (or agreed to participate). This can also be called “recruitment” rate.

Provide completion rate (Ratio agreed to participate/finished survey) if possible: The number of people agreeing to participate (or submitting the first survey page) divided by the number of people submitting the last questionnaire page. This is only relevant if there is a separate “informed consent” page or if the survey goes over several pages. This is a measure for attrition. Note that “completion” can involve leaving questionnaire items blank. This is not a measure for how completely questionnaires were filled in. (If you need a measure for this, use the word “completeness rate”.)

Were only completed questionnaires analyzed? Were questionnaires which terminated early (where, for example, users did not go through all questionnaire pages) also analyzed?

Reviewers' comments:

Reviewer's Responses to Questions

**Comments to the Author**

1. Is the manuscript technically sound, and do the data support the conclusions?

Reviewer #1: Yes

Reviewer #2: Yes

2. Has the statistical analysis been performed appropriately and rigorously? 

Reviewer #1: I Don't Know

Reviewer #2: Yes

3. Have the authors made all data underlying the findings in their manuscript fully available?

Reviewer #1: No

Reviewer #2: Yes

4. Is the manuscript presented in an intelligible fashion and written in standard English?

Reviewer #1: Yes

Reviewer #2: Yes

5. Review Comments to the Author

Reviewer #1: This is an interesting study that I believe would be of interest to the international community involved in developing and implementing the use of emergency care and treatment plans.

Whilst the paper has been presented well, I believe that some revision would improve it and increase its value to readers.

I feel that the title of the paper (in particular the use of the word ‘de-coding’) does not reflect the study’s findings and paper’s content accurately in a way that will help readers to find it and know what it is about. The main message from the findings of this study is that use of ambiguous language in resuscitation plans – unsurprisingly – results in different interpretation among doctors and in different decisions about choices of emergency treatments. I suggest that the title is amended to reflect this clearly.

The text states that 3 of the 4 authors ‘contributed equally to this work’. The nature and extent of their equal contributions are not stated. I suggest that this be corrected.

There is a degree of irony in this paper that addresses ambiguity and misinterpretation of terminology within a document that itself has the ambiguous name ‘resuscitation plan’. Some people use or interpret ‘resuscitation’ as referring to cardiopulmonary resuscitation, whereas others will use terminology such as ‘fluid resuscitation’ and some consider other emergency treatments as being part of ‘resuscitation’. The study identifies that doctors have different perceptions of the purpose and value of a ‘resuscitation plan’. The paper could usefully discuss whether that might be resolved, at least in part, by using a different, more specific name for the plan and/or a clear statement on the form of its purpose.

A weakness of this study, which should be acknowledged in the text, is that it has asked doctors to interpret (in the context of case vignettes) free-text phrases used on resuscitation plans from 2017 that were examined in a previous study. That previous study concluded that these phrases were ‘open to broad, variable interpretation’ and that ‘…ongoing education is integral to changing practice’. The present study has – unsurprisingly – validated the first of these conclusions, and – disappointingly – the current paper fails to acknowledge the crucial role of education (including audit, appraisal and feedback) in persuading doctors to record clear recommendations and stop using ambiguous phrases. If these phrases are in widespread use, that appears to indicate failure of adequate staff education in the completion of the 7-step form.

In more than one place, the text states ‘Terminology was neither consistently interpreted nor applied…’. I think that the authors intended to state that terminology was neither interpreted consistently nor applied consistently, not that the ‘terminology’ was not applied.

The paper refers to ‘not for Cardiopulmonary Resuscitation orders’ but does not make clear the legal status of such documents in Australia. In the UK, such documents are not legally binding ‘orders’ but are recommendations to guide those dealing with a sudden emergency to make the best possible decisions about the emergency care and treatment of the person, when the person has lost capacity to make choices about their care and treatment. The legal status of similar documents varies from country to country, so I think – especially in an international journal – that this should be acknowledged and discussed in this paper.

The paper states that the 7-step form is being used in all ‘public tertiary hospitals’ in South Australia. This terminology is potentially ambiguous for an international readership, so it would be helpful for the text to define the type, size and number of hospitals covered by this term.

It would be helpful and of great interest also to understand what arrangements are in place to communicate recommendations for the emergency care and treatment of patients who have not been admitted to this type of hospital, or patients transferred or discharged from a tertiary hospital. For the provision of person-centred care, I think there should be a robust process to transfer an up-to-date and relevant 7-step form with the patient or to generate current recommendations for emergency care and treatment on a different document. Whichever document is used outside the tertiary hospitals must be widely used, recognised and accepted across a range of other care settings, by a broad range of health and care professionals. Clearly, the ambiguous terminology used in this study to explore interpretation by hospital doctors would be likely to be even more ambiguous to health and care professionals if used on a form held by a patient at home or in various other care settings.

The paper implies in a single sentence that the 7-step form is very similar to 'international counterparts'. This is not strictly true. There are substantial differences, and it would be appropriate to identify these in the text and to consider them in subsequent discussion of possible options for improving the design of the 7-step form to overcome the problems identified in this study.

There is published evidence that using a discussion process and form that focus on and record treatments that would be wanted and should be considered to help to achieve the person’s goals of care, not primarily on treatment to be withheld, are preferred by both patients and by healthcare professionals. The ‘7-step form’ appears to focus more on withholding treatment rather than on giving realistic care and treatment to meet the person’s goals of care. As such, it does not appear truly person-centred or evidence-based.

In the published report of their previous study (reference 4) the authors concluded that the current Alert Form (‘7-step form’) could be improved. The tick-box list in section 4 of the form focuses firstly on emergency treatments to be withheld, emphasised using red print. Treatments that ‘will be provided’ are relegated to a rather less prominent free-text box below. Even the wording in this box is ambiguous. This box should list treatments that would be wanted and should be considered – and provided if they are needed. I would like to see much more discussion about the substantial potential to improve the form and to try to ensure that the form is used to support good communication, good medical practice and person-centred care, rather than primarily supporting the healthcare system.

In the discussion section, the paper suggests ‘To further improve clarity for junior staff, free-text options could be replaced with standardised choices e.g. non-invasive ventilation, cardiac monitoring, artificial hydration and intravenous antibiotics’. I am concerned that this would be primarily a system-focused intervention that is not truly person-centred. There are several potential ways in which the form could be modified to make it more person-centred and less likely to lead to varying interpretation and incorrect decision making in an emergency, and I would like to see the discussion expanded to consider these other options and the advantages and disadvantages of each. Trying to create a comprehensive list of tick-box treatment options to cater for every person's individual needs seems likely to make the form unduly long and cumbersome for all, and trying to limit the number of treatment options will inevitably fail to cater for all individual needs.

I am sorry to raise so many suggestions and hope that they will be viewed in the constructive spirit intended, and used to develop this interesting paper to its full potential.

Reviewer #2: I think you did a good job with a difficult research topic. I added a couple of comments about discussing the vignette management differences seen. I thought you addressed the language ambiguity in the second part well.

6. PLOS authors have the option to publish the peer review history of their article (what does this mean?). If published, this will include your full peer review and any attached files.

Reviewer #1: Yes: Dr David Pitcher

Reviewer #2: Yes: Michael Ritchie

---

## [Author Response · Author response to Decision Letter 0]

29 Oct 2019

Dear Reviewers,

Thank you for your considered and insightful feedback for this article, now titled "The impact of language on the interpretation of resuscitation clinical care plans by doctors. A mixed methods study." 

I have responded to your specific concerns in full within the 'Response to reviewers' letter. I have copied my responses in the area below as well. Overall I found your feedback highly valuable and feel the resubmitted article is significantly strengthened from the editorial and reviewer critique. I look forward to your responses.

Yours Sincerely,

Colette Dignam

Editorial Comments:

1) Review style requirements- I have reviewed the style requirements and made changes where necessary

2) Details of patient consent. I have provided further details in both the article and the online submission format

3) Data availability: Given your concern, I have referred this issue directly to the approving ethics committee. They have advised a restriction on uploading data to a public depository due to concerns about compromising participant privacy and confidentiality given the single-centre nature of the study. Data collection preceded recent changes to the Australian Research Council guidelines on data sharing and as such, consent for public data sharing was not sought from participants. The ethics committee has advised making ‘Data available to all interested researchers upon request’ and can be contacted directly at Health.CALHNResearchethics@sa.gov.au. My contact was Jasmine Kumar.

4) Please ensure your work adheres to the following guideline (CHERRIES) and provide citation: I have reviewed the CHERRIES checklist and amended my methodology to reflect their reporting guidelines, including citation:

Reviewer Comments and Feedback

1) Title does not reflect study's findings:

I agree with this feedback. I have amended the title to: “The impact of language on the interpretation of resuscitation clinical care plans by doctors. A mixed methods study.” 

2) The nature and extent of the authors equal contributions are not stated: 

The online PLOS ONE submission included a page detailing the nature of contributions. Please let me know if further information is required. 

3) There is a degree of irony in this paper that addresses ambiguity and misinterpretation of terminology within a document that itself has the ambiguous name ‘resuscitation plan’. ...The paper could usefully discuss whether that might be resolved, at least in part, by using a different, more specific name for the plan and/or a clear statement on the form of its purpose.

Response:

Irony aside, the lack of common nomenclature is certainly an issue, particularly when attempting a thorough literature review. I have revised the opening introductory paragraph to address this: “Clear documentation of medical treatment goals on a dedicated form helps patients receive appropriate and desired care, particularly in the setting of unexpected deterioration. Despite its recognised importance, there is no internationally accepted title for this type of documentation; terms in use include goals of care plan, resuscitation clinical care plan (resuscitation plan), and emergency treatment plan [1-3]. For the purposes of this study, we define a resuscitation plan as a document written by a doctor responsible for a patient’s care which includes a directive regarding cardiopulmonary resuscitation (CPR), but also contains recommendations and/or limitations about other interventions, such as intubation or use of life-prolonging antibiotics. Documentation should record the goals of care following a discussion with the patient or substitute decision-maker (SDM) about the treatment preferences, informed by the doctor’s medical assessment of likely treatment outcomes [1,2].” 

4) A weakness of this study, which should be acknowledged in the text, is...the current paper fails to acknowledge the crucial role of education... 

Response: On a practical level, the data collection for this study began 4 months after the initial study data was collated, based on preliminary analysis of the first study’s findings. The aim was to demonstrate that undefined language IS affecting care in the first instance (which, until now, was a theoretical concern). Designing and carrying out an educational campaign in this time frame was not possible. I have acknowledged this in limitations. 

“In addition to on-site education at the time of its launch at each new site, training about the 7-Step Pathway is available through online educational modules. The adequacy of these interventions to support the completion and interpretation of the form is a confounding factor that has not been addressed in this study. The capacity for a renewed education strategy and dedicated resources to improve the process, as opposed to changing the form, remains an area for future research.” 

5) I think that the authors intended to state that terminology was neither interpreted consistently nor applied consistently, not that the ‘terminology’ was not applied.

Response: The wording has been changed throughout the paper to reflect this clearer suggested phrasing. 

6) The paper refers to ‘not for Cardiopulmonary Resuscitation orders’ but does not make clear the legal status of such documents in Australia...

Response: The introductory text has been changed to reflect the relationship between legally binding Advance Care Directives and the 7-Step Pathway: 

“In 2014, the 7-Step Pathway Resuscitation Plan and Alert form (7-Step form) (S1 Fig 1) was introduced as an integral part of implementing the Advance Care Directives Act 2013 (SA) to provide a clinical tool to reinforce the principle of self-determination; the individual’s right to refuse treatment including CPR.”

“…While the 7-Step form is not legally binding in isolation, it may be used to reflect previously documented binding treatment refusals that a patient has recorded on an Advance Care Directive. “ 

7) The paper states that the 7-step form is being used in all ‘public tertiary hospitals’ in South Australia. This terminology is potentially ambiguous for an international readership, so it would be helpful for the text to define the type, size and number of hospitals covered by this term.

See next response

8) It would be helpful and of great interest also to understand what arrangements are in place to communicate recommendations for the emergency care and treatment of patients who have not been admitted to this type of hospital, or patients transferred or discharged from a tertiary hospital. For the provision of person-centred care, I think there should be a robust process to transfer an up-to-date and relevant 7-step form with the patient or to generate current recommendations for emergency care and treatment on a different document. Whichever document is used outside the tertiary hospitals must be widely used, recognised and accepted across a range of other care settings, by a broad range of health and care professionals. Clearly, the ambiguous terminology used in this study to explore interpretation by hospital doctors would be likely to be even more ambiguous to health and care professionals if used on a form held by a patient at home or in various other care settings.

Response: “This resuscitation plan is now used in all six South Australian urban public referral hospitals and is available in smaller rural hospitals and clinics staffed by primary care providers, it is also recognised as a treatment guide by ambulance officers [3]… The plan should be communicated to the primary care provider on hospital discharge, and a carbon-copy provided to the patient.”

9) The paper implies in a single sentence that the 7-step form is very similar to 'international counterparts'. This is not strictly true. There are substantial differences, and it would be appropriate to identify these in the text and to consider them in subsequent discussion of possible options for improving the design of the 7-step form..

Response: In addition to the Discussion, I have added the following to Introduction: “In line with this study’s definition, international examples of resuscitation plans also contain a CPR directive, a description of treatment limitations or recommendations, and a record of patient/SDM involvement. Key differences between examples include the relative focus on free text documentation vs standardised tick box care options, whether the document is used for a specified length of time or indefinitely (as is the case with the 7-Step form), and the involvement of senior staff in creating a plan [1,5,6].”

10) There is published evidence that using a discussion process and form that focus on and record treatments that would be wanted and should be considered to help to achieve the person’s goals of care, not primarily on treatment to be withheld, are preferred by both patients and by healthcare professionals. The ‘7-step form’ appears to focus more on withholding treatment rather than on giving realistic care and treatment to meet the person’s goals of care...

Response: The government-endorsed evidence-based approach that underpins form completion is perhaps under-represented in the manuscript. I have added the following to Introduction: “The form and underlying Resuscitation Planning Policy Framework were developed to address previously identified inadequacies in the process of resuscitation decisions and documentation, with a renewed emphasis on transparent patient-doctor discussion that would result in desired and appropriate medical care, while protecting the right to withhold futile interventions." 

"The completed form represents the final step in a process of creating a care plan in conjunction with the patient; a stepwise approach that is locally known as the 7-Step Pathway. In accordance with current evidence base, the 7-Step Pathway includes medical assessment, open consultation with the patient about their treatment goals, and a transparent discussion of appropriate treatment within the framework of known co-morbidity and quality of life [3,4]. ”

11) In the published report of their previous study (reference 4) the authors concluded that the current Alert Form (‘7-step form’) could be improved. The tick-box list in section 4 of the form focuses firstly on emergency treatments to be withheld, emphasised using red print. Treatments that ‘will be provided’ are relegated to a rather less prominent free-text box below. Even the wording in this box is ambiguous. This box should list treatments that would be wanted and should be considered – and provided if they are needed. I would like to see much more discussion about the substantial potential to improve the form and to try to ensure that the form is used to support good communication, good medical practice and person-centred care, rather than primarily supporting the healthcare system.

Response: I have addressed this and the following comment with a further paragraph in the Discussion; 

An ideal resuscitation plan therefore needs to strike a balance between patient advocacy, a clear medical handover, and the unpredictable nature of future hospital admissions. While junior doctors in this study were in favour of increased standardised choices such as ‘Not for non-invasive ventilation’, a list of this nature will inevitably be incomplete and maintains a focus on the treatment that will be withheld. Contemporary alternatives include the model exemplified in the USA-based POLST (physician’s orders for life sustaining treatment) model, where patients are divided into three broad treatment approaches; full, limited and palliative. The increased clarity and ease of use in this three-tiered care model potentially comes at the expense of a nuanced and individualised patient-centred approach. The ReSPECT (Recommended Summary Plan for Emergency Care and Treatment) model in United Kingdom provides ample scope for individualisation and patient involvement in the form of visual scales balancing curative and symptom-based care. This form requires the doctor to accurately document treatment limitations in free-text format however, thereby risking the use of undefined or ambiguous terminology that proved so problematic in this study. 

12) In the discussion section, the paper suggests ‘To further improve clarity for junior staff, free-text options could be replaced with standardised choices ...Trying to create a comprehensive list of tick-box treatment options to cater for every person's individual needs seems likely to make the form unduly long and cumbersome for all, and trying to limit the number of treatment options will inevitably fail to cater for all individual needs.

As above

Reviewer #2: I think you did a good job with a difficult research topic. I added a couple of comments about discussing the vignette management differences seen. I thought you addressed the language ambiguity in the second part well.

Response: I have expanded on your comments discussing the case vignette in the discussion: “Interns, who are inexperienced in clinical decision-making and are used to following senior orders, were also more likely to both closely follow a resuscitation plan and consider it a useful decision-making tool. This may in part explain the statistical differences in treatment approaches seen in the case vignettes. Internationally, many resuscitation plans require a secondary signature from a senior officer, which may be one practical solution to improve cohesion in the approaches of junior and senior staff.

The suggested grammatical and phrasing amendments have also been adopted throughout the paper

---

## [Editor Report · Decision Letter 1]

4 Nov 2019

The impact of language on the interpretation of resuscitation clinical care plans by doctors. A mixed methods study

PONE-D-19-20806R1

Dear Dr. Dignam,

We are pleased to inform you that your manuscript has been judged scientifically suitable for publication and will be formally accepted for publication once it complies with all outstanding technical requirements.

With kind regards,

Andrew Carl Miller

Academic Editor

PLOS ONE

Additional Editor Comments (optional):

The authors have satisfactorily answered the reviews queries.

---

## [Editor Report · Acceptance letter]

11 Nov 2019

PONE-D-19-20806R1 

The impact of language on the interpretation of resuscitation clinical care plans by doctors. A mixed methods study. 

Dear Dr. Dignam:

I am pleased to inform you that your manuscript has been deemed suitable for publication in PLOS ONE. Congratulations! Your manuscript is now with our production department. 

With kind regards,

on behalf of

Dr. Andrew Carl Miller 

Academic Editor

PLOS ONE